# InfoCL: Alleviating Catastrophic Forgetting in Continual Text Classification from An Information Theoretic Perspective

**Yifan Song**[1]  **Peiyi Wang**[1]  **Weimin Xiong**[1]  **Dawei Zhu**[1]
**Tianyu Liu**[2]  **Zhifang Sui**[1]  **Sujian Li**[1*]

[1]National Key Laboratory for Multimedia Information Processing,
School of Computer Science, Peking University
[2]Alibaba Group

{yfsong,wmxiong,dwzhu,tianyu0421,szf,lisujian}@pku.edu.cn
wangpeiyi9979@gmail.com

## Abstract

Continual learning (CL) aims to constantly learn new knowledge over time while avoiding catastrophic forgetting on old tasks. We focus on continual text classification under the class-incremental setting. Recent CL studies have identified the severe performance decrease on analogous classes as a key factor for catastrophic forgetting. In this paper, through an in-depth exploration of the representation learning process in CL, we discover that the compression effect of the information bottleneck leads to confusion on analogous classes. To enable the model learn more sufficient representations, we propose a novel replay-based continual text classification method, InfoCL. Our approach utilizes fast-slow and current-past contrastive learning to perform mutual information maximization and better recover the previously learned representations. In addition, InfoCL incorporates an adversarial memory augmentation strategy to alleviate the overfitting problem of replay. Experimental results demonstrate that InfoCL effectively mitigates forgetting and achieves state-of-the-art performance on three text classification tasks. The code is publicly available at https://github.com/Yifan-Song793/InfoCL.

## 1 Introduction

Continual learning (CL) enables conventional static natural language processing models to constantly gain new knowledge from a stream of incoming data (Sun et al., 2020; Biesialska et al., 2020). In this paper, we focus on continual text classification, which is formulated as a class-incremental problem, requiring the model to learn from a sequence of class-incremental tasks (Huang et al., 2021). Figure 1 gives an illustrative example of continual text classification. The model needs to learn to distinguish some new classes in each task

---

[*]Corresponding Author

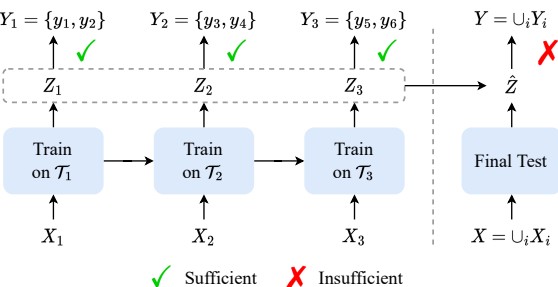

Figure 1: Illustration for continual text classification with three tasks where each task involves two new classes. $X_i$, $Y_i$, and $Z_i$ denote input sentences, new classes and learned representations for $i$-th task $\mathcal{T}_i$ respectively. Although the representations $Z_i$ learned in $\mathcal{T}_i$ are sufficient for classifying $Y_i$, they are insufficient to distinguish all seen classes $Y$ in the final test.

and is eventually evaluated on all seen classes. Like other CL systems, the major challenge of continual text classification is catastrophic forgetting: after new tasks are learned, performance on old tasks may degrade dramatically (Lange et al., 2022).

The earlier work in the CL community mainly attributes catastrophic forgetting to the corruption of the learned representations as new tasks arrive and various methods have been introduced to retain or recover previously learned representations (Kirkpatrick et al., 2017; Rebuffi et al., 2017; Mallya and Lazebnik, 2018; Lange et al., 2022). Recently, some studies (Wang et al., 2022; Zhao et al., 2023) find that, under the class-incremental setting, the severe performance decay among analogous classes is the key factor of catastrophic forgetting. To improve the performance of distinguishing analogous classes, Wang et al. (2022) exploit a heuristic adversarial class augmentation and Zhao et al. (2023) propose a sophisticated memory-insensitive prototype mechanism. However, due to a lack of thorough investigation into the underlying cause of confusion in similar classes, previous empirical methods may not be universally effective and are

unable to offer guidance for further improvements.

In this paper, for the first time we present an in-depth analysis of the analogous class confusion problem from an information theoretic perspective. We investigate the impact of the information bottleneck (IB) on the representation learning process of CL, specifically how it compresses the mutual information between the representation and the input. Through formal analysis and empirical study, we find that within each task, current CL models tend to discard features irrelevant to current task due to the compression effect of IB. While the acquired representations are locally sufficient for current task, they may be globally insufficient for classifying analogous classes in the final test. We refer to this phenomenon as **representation bias** and aim to enhance the CL model's ability to learn more comprehensive representations.

Based on our analysis, we propose a replay-based continual text classification method InfoCL. Using contrastive learning as the core mechanism, we enable the model to learn more comprehensive representations by maximizing the mutual information between representation and the original input via InfoNCE. Specifically, we design fast-slow and current-past contrast strategies. First, from the IB theory, the representations in the early stage of optimization preserves more information. Hence, when learning for new classes in current task, we leverage MoCo framework (He et al., 2020) and conduct fast-slow contrastive learning to facilitate the learned representations to retain more information about the input. On the other hand, to further alleviate representation corruption, when conducting memory replay, we leverage current-past contrastive learning to ensure the learned representations do not undergo significant changes. Due to the limited budget of memory, the performance of current-past contrastive learning is hindered by the over-fitting problem. To this end, InfoCL incorporates adversarial data augmentation to generate more training instances for replay.

Our contributions are summarized as follows: (1) We formally analyze the analogous class confusion problem in CL from an information theoretic perspective and derive that the representation bias led by the compression effect of IB is the underlying cause of forgetting. (2) We propose a novel replay-based continual text classification method InfoCL, which exploits fast-slow and current-past constrastive learning to capture more comprehen-

sive representations. (3) Experimental results on several text classification datasets show that InfoCL learns more effective representations and outperforms state-of-the-art methods.

## 2   Related Work

**Continual Learning**   Continual Learning (CL) studies the problem of continually learning knowledge from a sequence of tasks (Lange et al., 2022) while avoiding catastrophic forgetting. Previous CL work mainly attributes catastrophic forgetting to the corruption of learned knowledge and can be divided into three major families. *Replay-based* methods (Rebuffi et al., 2017; Prabhu et al., 2020) save a few previous task instances in a memory module and retrain on them while training new tasks. *Regularization-based* methods (Kirkpatrick et al., 2017; Aljundi et al., 2018) introduce an extra regularization loss to consolidate previous knowledge. *Parameter-isolation* methods (Mallya and Lazebnik, 2018) dynamically expand the network and dedicate different model parameters to each task. Recent studies have identified the confusion among analogous classes as a key factor of catastrophic forgetting. In this paper, we discover the representation bias is the underlying cause of such confusion and design InfoCL to mitigate it.

**Contrastive Learning**   Contrastive learning aims to learn representations by contrasting positive pairs against negative pairs (Chen et al., 2020). Recently, contrastive learning has made great progress in both unsupervised and supervised settings (Chen et al., 2020; He et al., 2020; Khosla et al., 2020; Barbano et al., 2022). The success of contrastive learning can be partially attributed to that the commonly used objective, InfoNCE, maximizes the mutual information between representations and inputs (van den Oord et al., 2018). Previous continual learning work has already integrated contrastive learning to alleviate the catastrophic forgetting. Cha et al. (2021) and Zhao et al. (2022) use supervised contrastive learning to learn more consistent representations. Hu et al. (2022) design a prototypical contrastive network to alleviate catastrophic forgetting. However, due to the lack of in-depth analysis of the representations learned in continual learning, these approaches fail to harness the full potential of contrastive learning[1]. In this paper, we investigate the representation learning

---

[1] See the Appendix A for detailed discussion of related work.

process in CL and propose fast-slow and current-past contrastive learning to enable the model learn more comprehensive representations and further mitigate the representation corruption problem.

# 3 Task Formulation

In this work, we focus on continual learning for a sequence of $k$ class-incremental text classification tasks $(\mathcal{T}_1, \mathcal{T}_2, ..., \mathcal{T}_k)$. Each task $\mathcal{T}_i$ has its dataset $\mathcal{D}_i = \{(x_n, y_n)\}_{n=1}^{N_i}$, where $(x_n, y_n)$ is an instance of current task and is sampled from an individually i.i.d. distribution $p(X_i, Y_i)$. Different tasks $\mathcal{T}_i$ and $\mathcal{T}_j$ have disjoint label sets $Y_i$ and $Y_j$. The goal of CL is to continually train the model on new tasks to learn new classes while avoiding forgetting previously learned ones. From another perspective, if we denote $X = \cup_i X_i$ and $Y = \cup_i Y_i$ as the input and output space of the entire CL process respectively, continual learning aims to approximate a holistic distribution $p(Y|X)$ from a non-i.i.d data stream.

The text classification model $F$ is usually composed of two modules: the encoder $f$ and the classifier $\sigma$. For an input $x$, we get the corresponding representation $\mathbf{z} = f(x)$, and use the logits $\sigma(\mathbf{z})$ to compute loss and predict the label.

# 4 Representation Bias in CL

Previous work (Wang et al., 2022; Zhao et al., 2023) reveals that the severe performance degradation on analogous classes is the key factor of catastrophic forgetting. In this section, we investigate the representation learning process of continual learning from an information theoretic perspective and find that the representation bias is the underlying cause of confusion in analogous classes.

## 4.1 Information Bottleneck

We first briefly introduce the background of information bottleneck in this section. Information bottleneck formulates the goal of deep learning as an information-theoretic trade-off between representation compression and preservation (Tishby and Zaslavsky, 2015; Shwartz-Ziv and Tishby, 2017). Given the input $\mathcal{X}$ and the label set $\mathcal{Y}$, one model is built to learn the representation $\mathcal{Z} = \mathcal{F}(\mathcal{X})$, where $\mathcal{F}$ is the encoder. The learning procedure of the model is to minimize the following Lagrangian:

$$I(\mathcal{X}; \mathcal{Z}) - \beta I(\mathcal{Z}; \mathcal{Y}), \tag{1}$$

where $I(\mathcal{X}; \mathcal{Z})$ is the mutual information (MI) between $\mathcal{X}$ and $\mathcal{Z}$, quantifying the information retained in the representation $\mathcal{Z}$. $I(\mathcal{Z}; \mathcal{Y})$ quantifies the amount of information in $\mathcal{Z}$ that enables the identification of the label $\mathcal{Y}$. $\beta$ is a trade-off hyperparameter. With information bottleneck, the model will learn *minimal sufficient representation* $\mathcal{Z}^*$ (Achille and Soatto, 2018) of $\mathcal{X}$ corresponding to $\mathcal{Y}$:

$$\mathcal{Z}^* = \arg \min_{\mathcal{Z}} I(\mathcal{X}; \mathcal{Z}) \tag{2}$$

$$\text{s.t. } I(\mathcal{Z}; \mathcal{Y}) = I(\mathcal{X}; \mathcal{Y}). \tag{3}$$

Minimal sufficient representation is important for supervised learning, because it retains as little about input as possible to simplify the role of the classifier and improve generalization, without losing information about labels.

## 4.2 Representation Learning Process of CL

Continual learning is formulated as a sequence of individual tasks $(\mathcal{T}_1, \mathcal{T}_2, ..., \mathcal{T}_k)$. For $i$-th task $\mathcal{T}_i$, the model aims to approximate distribution of current task $p(Y_i|X_i)$. According to IB, if the model $F = \sigma \circ f$ converges, the learned hidden representation $Z_i = f(X_i)$ will be local minimal sufficient for $\mathcal{T}_i$:

$$Z_i = \arg \min_{Z_i} I(X_i; Z_i) \tag{4}$$

$$\text{s.t. } I(Z_i; Y_i) = I(X_i; Y_i), \tag{5}$$

which ensures the performance and generalization ability of the current task. Nevertheless, the local minimization of the compression term $I(X_i; Z_i)$ will bring potential risks: features that are useless in the current task but crucial for other tasks will be discarded.

The goal of CL is to classify all seen classes $Y = \cup_i Y_i$. For the entire continual learning task with the holistic target distribution $p(Y|X)$, the necessary condition to perform well is that the representation $Z$ is globally sufficient for $Y$: $I(Z; Y) = I(X; Y)$. However, as some crucial features are compressed, the combination of local minimal sufficient representations for each task $Z = \cup_i Z_i$ may be globally insufficient:

$$I(Z; Y) < I(X; Y). \tag{6}$$

We name this phenomenon as **representation bias**: due to the compression effect of IB, the learned

| Models | FewRel | | MAVEN | |
|---|---|---|---|---|
| | $I(X_1; Z_1)$ | $I(Z; Y)$ | $I(X_1; Z_1)$ | $I(Z; Y)$ |
| Supervised | 2.42 | 2.45 | 3.50 | 2.42 |
| RP-CRE | 2.08 | 2.21 | 3.15 | 2.31 |
| CRL | 2.12 | 2.18 | 3.12 | 2.30 |
| CRECL | 2.20 | 2.31 | 3.01 | 2.36 |

Table 1: Mutual information comparison between supervised learning and strong CL baselines on FewRel and MAVEN datasets. We use $I(X; Z)$ to measure how much features of input $X$ representation $Z$ preserves. To exclude the impact of representation corruption, we instead estimate $I(X_1; Z_1)$ after CL models finish $\mathcal{T}_1$. $I(Z; Y)$ measures whether the learned representation is sufficient for the entire continual task.

representations in each individual task may be insufficient for the entire continual task.

Then the underlying cause of the performance decrease of analogous classes is obvious. Take two cross-task analogous classes $y_a$ and $y_b$ as an example. Under sequential task setting of CL, the model is unable to co-training on instances from $y_a$ and $y_b$. It means that the representations of these two classes can exclusively be learned within their respective tasks. When learning $y_a$, the local sufficient representations to identify $y_a$ are insufficient to differentiate with $y_b$. Hence, the appearance of $y_b$ will lead to a dramatically performance decrease of $y_a$, resulting in what is known as catastrophic forgetting.

### 4.3 Empirical Results

To confirm our analysis, here we directly measure the mutual information among $X$, $Y$ and $Z$. Since the representations learned by supervised learning is always globally sufficient, i.e., $I(Z; Y) = I(X; Y)$, we use supervised learning on all data as the baseline, and compare it with several strong CL methods. Concretely, we use MINE (Belghazi et al., 2018) as the MI estimator and conduct experiments on FewRel and MAVEN datasets[2].

First, we measure $I(X; Z)$ to quantify the features preserved in the representation $Z$. However, previously learned representations will be corrupted once the model learns new tasks, leading to inaccurate estimation. To exclude the impact of representation corruption, we instead estimate $I(X_1; Z_1)$ on $\mathcal{T}_1$'s test set. Second, to assess whether learned representations are sufficient for

---

[2]See Section 6.1 for details of CL baselines and datasets.

the entire continual task, we compare $I(Z; Y)$ on the final test set with all classes.

As shown in Table 1, both $I(X_1; Z_1)$ and $I(Z; Y)$ of three CL models are significantly lower than supervised learning, indicating that the CL model tends to compress more information due to the individual task setting and the representations learned in CL are insufficient for the entire continual task.

## 5 Methodology

From the formal analysis and empirical verification, we establish that representation bias plays a crucial role in the performance decline of analogous classes. Consequently, in this section, we propose a novel replay-based CL method, InfoCL, which is able to maximize $I(X_i; Z_i)$ and help the model learn more comprehensive representations.

### 5.1 Overall Framework

The objective of contrastive learning, specifically InfoNCE, serves as a proxy to maximize the mutual information $I(X_i; Z_i)$ (van den Oord et al., 2018). Therefore, we utilize contrastive learning as the core mechanism of InfoCL to acquire more comprehensive representations. Concretely, we design fast-slow and current-past contrastive learning for training new data and memory data.

The overall framework of InfoCL is depicted in Figure 2. For a new task $\mathcal{T}_k$, we first train the model on $\mathcal{D}_k$ to learn this task. We perform fast-slow contrastive learning to help the model capture more sufficient representations and mitigate the representation bias. Then we store a few typical instances for each class $y \in Y_k$ into the memory $\mathcal{M}$, which contains instances of all seen classes. To alleviate representation decay, we next conduct memory replay with current-past contrastive learning. As the performance of representation recovery is always hindered by the limited size of memory, we incorporates adversarial augmentation to alleviate overfitting.

### 5.2 Fast-Slow Contrastive Learning

In the representation learning process of a task $\mathcal{T}_i$, the compression effect of IB will minimize the mutual information $I(X_i; Z_i)$, leading to globally insufficient representations. Intuitively, in the early phase of optimization, $I(X_i; Z_i)$ is larger and the representations preserve more information about the inputs. However, directly adopting an early

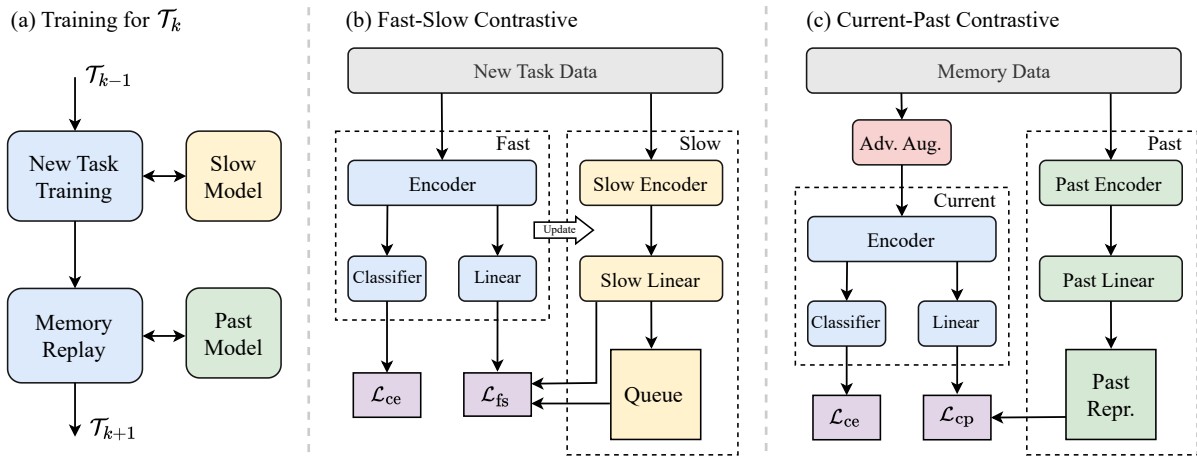

Figure 2: (a) A demonstration for InfoCL. We design fast-slow and current-past contrastive learning for initial training and memory replay, respectively. (b) Fast-slow contrastive learning. The slowly progressing model generates representations preserving more information. (c) Current-past contrastive learning with adversarial augmentation. Contrasting with old model from $\mathcal{T}_{k-1}$ further alleviates representation corruption.

stop strategy is not feasible, as the representation compression is essential for generalization. Instead, we try to pull the learned representations and early representations of the same class together, facilitating the preservation of more comprehensive information in the final representations. Inspired by He et al. (2020), we employ a momentum contrast which consists of a fast encoder and a momentum updated slow encoder. The representations from the slowly-updated branch will preserve more information about the input sentences. The fast-slow contrast can "distill" these information from the slow branch to fast branch to learn more comprehensive representations.

Figure 2 (b) depicts the architecture. The fast model is updated by gradient descent, while the slow model truncates the gradient and is updated with the momentum mechanism during training. Formally, denoting the parameters of the fast and slow models as $\theta$ and $\theta'$, $\theta'$ is updated by:

$$\theta' \leftarrow \eta\theta' + (1 - \eta)\theta, \qquad (7)$$

where $\eta$ is the momentum coefficient which is relatively large to ensure the slow update (e.g., $\eta = 0.99$). For the slow encoder, we also maintain a representation queue $Q$ to increase the number of negative instances beyond the batch size, enabling InfoNCE to more effectively maximize MI. The queue is updated with the output of slow model by first-in-first-out strategy.

We denote $\mathbf{z}$ for the representations from the fast model and $\widetilde{\mathbf{z}}$ for the slow model. Then the slow

representations $\widetilde{\mathbf{z}}$ preserve more information than $\mathbf{z}$. We use InfoNCE to perform fast-slow contrast:

$$\mathcal{L}_{\text{fs}} = -\frac{1}{|B|} \sum_{i \in I} \sum_{p \in P(i)} \log \frac{\exp(\mathbf{z}_i \cdot \widetilde{\mathbf{z}}_p / \tau_1)}{\sum_{j \in J} \exp(\mathbf{z}_i \cdot \widetilde{\mathbf{z}}_j / \tau_1)},$$

$$(8)$$

where $I = \{1, 2, ..., |B|\}$ is the set of indices of batch $B$. $J = \{1, 2, ..., |B \cup Q|\}$ denotes the indices set of instances in the batch or the queue. $P(i) = \{p \in J : y_p = y_i\}$ is the indices of instances which have the same label as $\mathbf{z}_i$ from the batch or the queue. $\tau_1$ is the temperature hyperparameter.

The final optimization objective in new task training is the combination of cross entropy loss $\mathcal{L}_{\text{ce}}$ and the contrastive loss $\mathcal{L}_{\text{fs}}$:

$$\mathcal{L}_1 = \mathcal{L}_{\text{ce}} + \lambda_1 \mathcal{L}_{\text{fs}}, \qquad (9)$$

where $\lambda_1$ is the factor to adjust the loss weight.

### 5.3 Memory Selection

After the initial training stage, we select and store typical instances for each class for replay. Since the primary focus of our paper is to address the representation bias problem, we adopt the memory sampling strategy employed in prior work (Cui et al., 2021; Zhao et al., 2022) to ensure a fair comparison. Specifically, for each class, we use K-means to cluster the corresponding representations, and the instances closest to the centroids are stored in memory $\mathcal{M}$. Then we use the instances

of all seen classes from memory $\mathcal{M}$ to conduct the memory replay stage.

## 5.4 Current-Past Contrastive Learning

When performing memory replay, we also employ contrastive learning to enable the model to learn more comprehensive representations for all previously seen classes. Additionally, to further enhance representation recovery in memory replay stage, we propose current-past contrastive learning which explicitly aligns current representations to the previous ones.

As shown in Figure 2 (c), after the model finishs $\mathcal{T}_{k-1}$, we store the representations $\bar{\mathbf{z}}$ of the instances from memory $\mathcal{M}$. Then we use InfoNCE loss to pull current representations $\mathbf{z}$ and past representations $\bar{\mathbf{z}}$ of the same class together:

$$\mathcal{L}_{\mathrm{cp}} = -\frac{1}{|B|} \sum_{i \in I} \sum_{p \in P(i)} \log \frac{\exp(\mathbf{z}_i \cdot \bar{\mathbf{z}}_p / \tau_2)}{\sum_{m \in M} \exp(\mathbf{z}_i \cdot \bar{\mathbf{z}}_m / \tau_2)},$$
(10)

where $I = \{1, 2, ..., |B|\}$ is the set of indices of batch $B$. $M = \{1, 2, ..., |\mathcal{M}|\}$ denotes the indices set of instances in memory $\mathcal{M}$. $P(i) = \{p \in M : y_p = y_i\}$ is the indices of instances which have the same label as $\mathbf{z}_i$ from the memory. $\tau_2$ is the temperature hyperparameter.

The optimization objective in the memory replay stage is

$$\mathcal{L}_2 = \mathcal{L}_{\mathrm{ce}} + \lambda_2 \mathcal{L}_{\mathrm{cp}},$$
(11)

where $\mathcal{L}_{\mathrm{ce}}$ is cross entropy loss and $\lambda_2$ is the factor to adjust the loss weight.

## 5.5 Adversarial Memory Augmentation

Due to the constrained memory budgets, the performance of current-past contrastive learning in the memory replay stage is hindered by the overfitting problem. To alleviate overfitting and enhance the effect of representation recovery, we incorporate adversarial data augmentation (Zhu et al., 2020):

$$\mathcal{L}_{\mathrm{adv}} =$$
$$\min_{\theta} \mathbb{E}_{(x,y) \sim \mathcal{M}} \left[ \frac{1}{K} \sum_{t=0}^{K-1} \max_{\|\delta_t\| \leq \epsilon} \mathcal{L}_2 \left( F(x + \delta_t), y \right) \right].$$
(12)

Intuitively, it performs multiple adversarial attack iterations to craft adversarial examples, which is equivalent to replacing the original batch with a $K$-times larger adversarial augmented batch. Please refer to Appendix B for details about Eq. 12.

## 6 Experiments

### 6.1 Experiment Setups

**Datasets**  To fully measure the ability of InfoCL, we conduct experiments on 4 datasets for 3 different text classification tasks, including relation extraction, event classification, and intent detection. For relation extraction, following previous work (Han et al., 2020; Cui et al., 2021; Zhao et al., 2022), we use **FewRel** (Han et al., 2018) and **TACRED** (Zhang et al., 2017). For event classification, following Yu et al. (2021) and Wu et al. (2022), we use **MAVEN** (Wang et al., 2020) to build our benchmark. For intent detection, following Liu et al. (2021), we choose **HWU64** (Liu et al., 2019) dataset. For the task sequence, we simulate 10 tasks by randomly dividing all classes of the dataset into 10 disjoint sets, and the number of new classes in each task for FewRel, TACRED, MAVEN and HWU64 are 8, 4, 12, 5 respectively. For a fair comparison, the result of baselines are reproduced on the same task sequences as our method. Please refer to Appendix C for details of these four datasets. Following previous work (Hu et al., 2022; Wang et al., 2022), we use the average accuracy (Acc) on all seen tasks as the metric.

**Baselines**  We compare InfoCL against the following baselines: IDBR (Huang et al., 2021), KCN (Cao et al., 2020), KDRK (Yu et al., 2021), EMAR (Han et al., 2020), RP-CRE (Cui et al., 2021), CRL (Zhao et al., 2022), CRECL (Hu et al., 2022), ACA (Wang et al., 2022) and CEAR (Zhao et al., 2023). See Appendix D for details of the baselines.

Some baselines are originally proposed to tackle one specific task. For example, RP-CRE is designed for continual relation extraction. We adapt these baselines to other tasks and report the corresponding results. Since ACA and CEAR consist of data augmentation specially designed for relation extraction, they cannot be adapted to other tasks.

**Implementation Details**  For InfoCL, we use $\text{BERT}_{\text{base}}$ (Devlin et al., 2019) as the encoder following previous work (Cui et al., 2021; Wang et al., 2022). The learning rate of InfoCL is set to 1e-5 for the BERT encoder and 1e-3 for other modules. Hyperparameters are tuned on the first three tasks. The memory budget for each class is fixed at 10 for all methods. For all experiments, we use NVIDIA A800 GPUs and report the average result of 5 different task sequences. More implementation details can be found in Appendix E.

| Datasets | FewRel | | | TACRED | | | MAVEN | | | HWU64 | | |
|---|---|---|---|---|---|---|---|---|---|---|---|---|
| Models | $\mathcal{T}_8$ | $\mathcal{T}_9$ | $\mathcal{T}_{10}$ | $\mathcal{T}_8$ | $\mathcal{T}_9$ | $\mathcal{T}_{10}$ | $\mathcal{T}_8$ | $\mathcal{T}_9$ | $\mathcal{T}_{10}$ | $\mathcal{T}_8$ | $\mathcal{T}_9$ | $\mathcal{T}_{10}$ |
| IDBR (Huang et al., 2021) | 73.7 | 71.7 | 68.9 | 64.2 | 63.8 | 60.1 | 64.4 | 60.2 | 57.3 | 80.2 | 78.0 | 76.2 |
| KCN (Cao et al., 2020) | 80.3 | 78.8 | 76.0 | 72.1 | 72.2 | 70.6 | 68.4 | 67.7 | 64.4 | 83.7 | 82.7 | 81.9 |
| KDRK (Yu et al., 2021) | 81.6 | 80.2 | 78.0 | 72.9 | 72.1 | 70.8 | 69.6 | 68.9 | 65.4 | 85.1 | 82.5 | 81.4 |
| EMAR (Han et al., 2020) | 86.1 | 84.8 | 83.6 | 76.6 | 76.8 | 76.1 | 76.8 | 75.7 | 73.2 | 85.5 | 83.9 | 83.1 |
| RP-CRE (Cui et al., 2021) | 85.8 | 84.4 | 82.8 | 76.1 | 75.0 | 75.3 | 77.1 | 76.0 | 73.6 | 84.5 | 83.8 | 82.7 |
| CRL (Zhao et al., 2022) | 85.6 | 84.5 | 83.1 | 79.1 | 79.0 | 78.0 | 76.8 | 75.9 | 73.7 | 83.1 | 81.3 | 81.5 |
| CRECL (Hu et al., 2022) | 84.6 | 83.6 | 82.7 | **81.4** | 79.3 | 78.5 | 75.9 | 75.1 | 73.5 | 83.1 | 81.9 | 81.1 |
| ACA (Wang et al., 2022) | 87.0 | 86.3 | 84.7 | 78.6 | 78.8 | 78.1 | – | – | – | – | – | – |
| CEAR (Zhao et al., 2023) | 86.9 | 85.6 | 84.2 | 81.1 | **80.1** | **79.1** | – | – | – | – | – | – |
| InfoCL (Ours) | **87.8** | **86.8** | **85.4** | 79.7 | 78.4 | 78.2 | **78.2** | **77.1** | **75.3** | **86.3** | **85.3** | **84.1** |

Table 2: Accuracy (%) on all seen classes after learning the last three tasks. We report the average result of 5 different runs. The best results are in **boldface**. ACA and CEAR is specially designed for continual relation extraction and cannot be adapted to other tasks.

| Models | Few. | TAC. | MAV. | HWU. |
|---|---|---|---|---|
| InfoCL | 85.4 | 78.2 | 75.3 | 84.1 |
| w/o f-s con. | 84.9 | 77.6 | 74.7 | 83.4 |
| w/o c-p con. | 85.0 | 78.1 | 75.1 | 83.7 |
| w/o adv. aug. | 84.8 | 77.9 | 75.0 | 83.6 |

Table 3: Ablation study of InfoCL. "f-s con." and "c-p con." denote fast-slow and current-past contrastive learning. "adv. aug." denotes adversarial memory augmentaion mechanism.

| Models | FewRel | | MAVEN | |
|---|---|---|---|---|
| | Accuracy | Drop | Accuracy | Drop |
| CRL | 75.3 | 13.3 | 59.8 | 21.2 |
| CRECL | 74.9 | 13.6 | 59.2 | 21.9 |
| InfoCL | **78.6** | **11.8** | **61.3** | **20.7** |

Table 4: Average Accuracy (%) and accuracy drop (%) on analogous classes. For each dataset, we select 20% classes which are most likely to be confused with other classes.

## 6.2 Main Results

Table 2 shows the performance of InfoCL and baselines on four datasets for three text classification tasks. Due to space constraints, we only illustrate results on the last three tasks. The complete accuracy and standard deviation of all 10 tasks can be found in Appendix G. As shown, on FewRel, MAVEN and HWU64, our proposed InfoCL consistently outperforms all baselines and achieves new state-of-the-art results. These experimental results demonstrate the effectiveness and universality of our proposed method. Regarding the TACRED dataset, it has been noticed that a large fraction of the examples are mislabeled, thus compromising the reliability of the evaluation (Alt et al., 2020; Stoica et al., 2021). Here we strongly advocate for a more reliable evaluation on other high-quality text classification datasets.

## 7 Analysis

### 7.1 Ablation Study

We conduct an ablation study to investigate the effectiveness of different components of InfoCL. The results are shown in Table 3. We find that the three core mechanisms of InfoCL, namely fast-slow contrast, current-past contrast, and adversarial memory augmentation, are conducive to the model performance. Furthermore, the fast-slow contrastive learning performed in the new task training stage seems to be more effective than the other components, indicating that learning comprehensive representations for the new task is more essential to mitigate representation bias problem.

### 7.2 Performance on Analogous Classes

We reveal that the representation bias problem is the underlying cause of confusion on analogous classes in CL. Since InfoCL aims to learn more sufficient representations and mitigate the bias, we also conduct experiment to explore this point. Following (Wang et al., 2022), we use the cosine distance of the average embedding of the instances as a metric to identify analogous classes. Specifically, we select 16 and 24 classes (20% of all of the classes) for FewRel and MAVEN, which are most likely to be confused with other classes. The list of these classes are shown in Appendix F. If

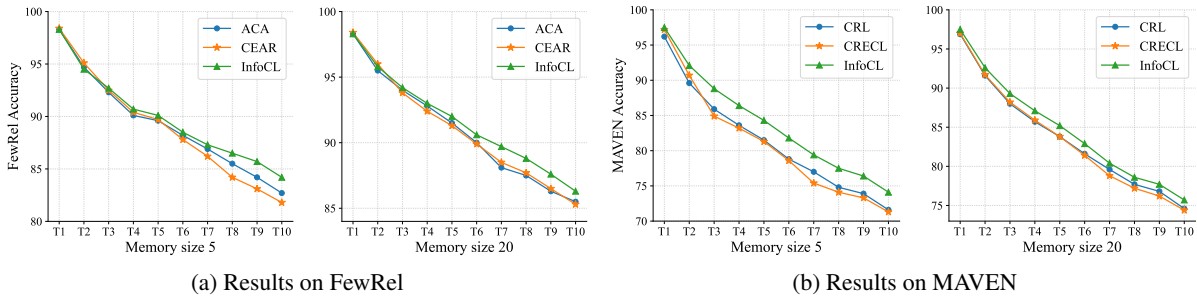

(a) Results on FewRel

(b) Results on MAVEN

Figure 3: Accuracy (%) w.r.t. different memory sizes of different methods.

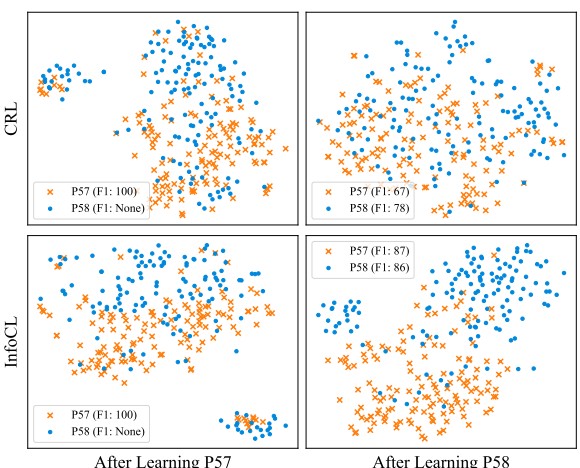

Figure 4: The representations of instances from P57 ("director") and P58 ("screenwriter"). P57 and P58 are in different tasks and P57 is learned before P58 appears.

these selected classes are in the former five tasks, we record the average final accuracy and the accuracy drop on them. As shown in Table 4, the performance on analogous classes of our model is superior and drops the least, demonstrating that our model succeeds in alleviating confusion among analogous classes.

### 7.3 Comprehensive Representation Learning

InfoCL employs contrastive learning to learn comprehensive representations via mutual information maximization. To assess the efficiency of representation learning process of our method, we directly compute the mutual information of the representations. Same as Section 4.3, we measure $I(X_1; Z_1)$ on $\mathcal{T}_1$ and $I(Z; Y)$ after the final task with all classes. For InfoCL, $I(X_1; Z_1) = 2.32$ and $I(Z; Y) = 2.38$ on FewRel, $I(X_1; Z_1) = 3.20$ and $I(Z; Y) = 2.37$ on MAVEN. Compared with baselines in Table 1, both mutual information metrics are higher, indicating our method can learn more sufficient representation to mitigate the

representation bias.

To provide an intuitive demonstration of the effective representations acquired by InfoCL, we conducted a case study. We select two analogous classes from FewRel, P57 ("director") and P58 ("screenwriter"). We use t-SNE to visualize the representations of these two classes after the model learned them. As illustrated in Figure 4, for both methods, the accuracy of P57 reaches 100% after learning it. However, due to the representation bias, when P58 appears, the accuracy of CRL dramatically declined. In contrast, InfoCL maintains a relatively stable performance. Notably, even without training for P58, the representations of two classes in our method exhibit a noticeable level of differentiation, highlighting InfoCL's capacity to acquire more comprehensive representations.

### 7.4 Influence of Memory Size

Memory size is the number of stored instances for each class, which is an important factor for the performance of replay-based CL methods. Therefore, in this section, we study the impact of memory size on InfoCL. As Table 2 has reported the results with memory size 10, here we compare the performance of InfoCL with strong baselines on FewRel and MAVEN under memory sizes 5 and 20.

Table 3 demonstrates that InfoCL consistently outperforms strong baselines across various memory sizes. Surprisingly, even with a memory size of 5, InfoCL achieves comparable performance to the baselines with a memory size of 20, highlighting its superior capability. As the memory size decreases, the performance of all models degrades, showing the importance of memory for replay-based methods. Whereas, InfoCL maintains a relatively stable performance, showcasing its robustness even in extreme scenarios.

# 8 Conclusion

In this paper, we focus on continual learning for text classification in the class-incremental setting. We formally investigate the representation learning process of CL and discover the representation bias will lead to catastrophic forgetting on analogous classes. Based on our analysis, we propose InfoCL, which utilizes fast-slow and current-past contrastive learning to learn more comprehensive representations and alleviate representation corruption. An adversarial augmentation strategy is also employed to further enhance the performance of the representation recovery. Experimental results show that InfoCL learns effective representations and outperforms the latest baselines.

## Limitations

Our paper has several limitations: (1) Our proposed InfoCL utilizes fast-slow and current-past contrastive learning to learn more comprehensive representations, which introduces extra computational overhead and is less efficient than other replay-based CL methods; (2) We only focus on catastrophic forgetting problem in continual text classification. How to encourage knowledge transfer in CL is not explored in this paper.

## Ethics Statement

Our work complies with the ACL Ethics Policy. Since text classification is a standard task in NLP and all datasets we used are publicly available, we have not identified any significant ethical considerations associated with our work.

## Acknowledgement

We thank the anonymous reviewers for their helpful comments on this paper. This work was partially supported by National Key R&D Program of China (No. 2022YFC3600402) and National Social Science Foundation Project of China (21&ZD287).

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

## A Comparison with Some Related Work

As aforementioned in Section 2, some related work also applied contrastive learning on continual learning. In this section, we provide more detailed discussion of these work.

Both of Co$^2$L (Cha et al., 2021) and CRL (Zhao et al., 2022) use supervised contrastive learning and knowledge distillation to learn more consistent representations. We would like to highlight two significant differences between our work and them. First, we formally analyze the representation bias problem from an information bottleneck perspective and propose that maximizing $I(X_i; Z_i)$ can effectively mitigate this bias. Second, while Co2L employs a vanilla supervised contrastive loss for representation learning, we introduce two novel contrastive learning techniques: fast-slow and current-past contrastive learning. Our method not only enables the model to acquire more comprehensive representations but also improves its ability to retain previously learned knowledge. As shown in Table 2, our method outperforms Co$^2$L/CRL on four datasets.

CLASSIC (Ke et al., 2021) utilizes a contrastive continual learning method for continual aspect sentiment classification task. However, it is specifically designed for the domain-incremental scenario, where its primary goal is to facilitate knowledge transfer across tasks. This objective is distinct from our class-incremental scenario, and thus, the use of InfoNCE in CLASSIC is not directly comparable to our work.

OCM (Guo et al., 2022) proposes mutual information maximization to learn holistic representation for online continual image classification. Our work, in comparison to OCM, shares a similar motivation of learning more comprehensive representations. Here we provide clarifications on several key differences between our work and OCM. First, OCM focuses on online continual learning on image classification, which is different with our scenario. Second, for the first time, we provide a formal analysis of catastrophic forgetting from the perspective of the information bottleneck. This analysis sets us apart from OCM. Finally, the contrastive learning in OCM relies on data augmentation of the image to ensure performance. According to their ablation study, without the random-resized-crop augmentation, the performance of OCM will drop drastically. In contrast, we design fast-slow and current-past contrastive learning to get better representations. This contrastive learning design is also a distinguishing feature of our work.

## B Details about Eq. 12

In memory replay stage, to alleviate the overfitting on memorized instances, we introduce adversarial data augmentation (Zhu et al., 2020):

$$\mathcal{L}_{\text{adv}} =$$
$$\min_\theta \mathbb{E}_{(x,y)\sim\mathcal{M}} \left[ \frac{1}{K} \sum_{t=0}^{K-1} \max_{\|\delta_t\|\le\epsilon} \mathcal{L}\left(F(x+\delta_t), y\right) \right],$$
(13)

where $F$ is the text classification model and $(x, y)$ is a batch of data from the memory bank $\mathcal{M}$, $\delta$ is the perturbation constrained within the $\epsilon$-ball, $K$ is step size hyperparameter. The inner maximization problem in (13) is to find the worst-case adversarial examples to maximize the training loss, while the outer minimization problem in (13) aims at optimizing the model to minimize the loss of adversarial examples. The inner maximization problem is solved iteratively:

$$\nabla(\delta_{t-1}) = \nabla_\delta \mathcal{L}\left(F(x+\delta_{t-1}), y\right), \quad (14)$$
$$\delta_t = \prod_{\|\delta\|\le\epsilon} \left( \delta_{t-1} + \alpha \cdot \frac{\nabla(\delta_{t-1})}{\|\nabla(\delta_{t-1})\|} \right), \quad (15)$$

where $\delta_t$ is the perturbation in $t$-th step and $\prod_{\|\delta\|\le\epsilon}(\cdot)$ projects the perturbation onto the $\epsilon$-ball, $\alpha$ is step size.

Intuitively, it performs multiple adversarial attack iterations to craft adversarial examples, and simultaneously accumulates the free parameter gradients $\nabla_\theta \mathcal{L}$ in each iteration. After that, the model parameter $\theta$ is updated all at once with the accumulated gradients, which is equivalent to replacing the original batch with a $K$-times larger adversarial augmented batch.

## C Dataset Details

**FewRel** (Han et al., 2018) It is a large scale relation extraction dataset containing 80 relations. FewRel is a balanced dataset and each relation has 700 instances. Following Zhao et al. (2022); Wang et al. (2022), we merge the original train and valid set of FewRel and for each relation we sample 420 instances for training and 140 instances for test. FewRel is licensed under MIT License.

**TACRED** (Zhang et al., 2017) It is a crowdsourcing relation extraction dataset containing 42 relations (including *no_relation*) and 106264 instances.

Following Zhao et al. (2022); Wang et al. (2022), we remove *no_relation* and in our experiments. Since TACRED is a imbalanced dataset, for each relation the number of training instances is limited to 320 and the number of test instances is limited to 40. TACRED is licensed under LDC User Agreement for Non-Members.

**MAVEN** (Wang et al., 2020) It is a large scale event detection dataset with 168 event types. Since MAVEN has a severe long-tail distribution, we use the data of the top 120 frequent classes. The original test set of MAVEN is not publicly available, and we use the original development set as our test set. MAVEN is licensed under Apache License 2.0.

**HWU64** (Liu et al., 2019) It is an intent classification dataset with 64 intent classes. Following Liu et al. (2021), we use the data of the top 50 frequent classes and the total number of instances are 24137. HWU64 is licensed under CC-BY-4.0 License.

## D   Baselines

**IDBR** (Huang et al., 2021) proposes an information disentanglement method to learn representations that can well generalize to future tasks. **KCN** (Cao et al., 2020) utilizes prototype retrospection and hierarchical distillation to consolidate knowledge. **KDRK** (Yu et al., 2021) encourages knowledge transfer between old and new classes. **EMAR** (Han et al., 2020) proposes a memory activation and reconsolidation mechanism to retain the learned knowledge. **RP-CRE** (Cui et al., 2021) proposes a memory network to retain the learned representations with class prototypes. **CRL** (Zhao et al., 2022) adopts contrastive learning replay and knowledge distillation to retain the learned knowledge. **CRECL** (Hu et al., 2022) uses a prototypical contrastive network to defy forgetting. **ACA** (Wang et al., 2022) designs two adversarial class augmentation mechanism to learn robust representations. **CEAR** (Zhao et al., 2023) proposes memory-intensive relation prototypes and memory augmentation to reduce overfitting to typical samples in rehearsal stage.

## E   Implementation Details

We implement InfoCL with PyTorch (Paszke et al., 2019) and HuggingFace Transformers (Wolf et al., 2020). Following previous work (Cui et al., 2021; Wang et al., 2022), we use BERT$_{base}$ (Devlin et al., 2019) as encoder. PyTorch is licensed under the

modified BSD license. HuggingFace Transformers and BERT$_{base}$ are licensed under the Apache License 2.0. Our use of existing artifacts is consistent with their intended use.

Specifically, for the input $x$ in relation extraction, we use $[E_{11}]$, $[E_{12}]$, $[E_{21}]$ and $[E_{22}]$ to denote the start and end position of head and tail entity respectively, and the representation $z$ is the concatenation of the last hidden states of $[E_{11}]$ and $[E_{21}]$. For the input $x$ in event detection, the representation $z$ is the average pooling of the last hidden states of the trigger words. For the input $x$ in intent detection, the representation $z$ is the average pooling of the last hidden states of the whole sentence.

The learning rate of InfoCL is set to 1e-5 for the BERT encoder and 1e-3 for the other modules. For FewRel, MAVEN and HWU64, the batch size is 32. For TACRED, the batch size is 16 because of the small amount of training data. The budget of memory bank for each class is 10 for all methods. The size of the queue $Q$ in fast-slow contrastive learning is 512 and the momentum coefficient $\eta$ is 0.99. The temperatures $\tau_1$, $\tau_2$ are set to 0.05. The loss factors $\lambda_1$, $\lambda_2$ are 0.05. For the adversarial memory augmentation, $K$=2, $\epsilon$=3e-1, $\alpha$=1e-1. For all experiments, we use NVIDIA A800 GPUs and report the average result of 5 different task sequences.

## F   Analogous Classes for Evaluation

We select the top 20% classes which are most likely to be confused with other classes from FewRel and MAVEN.

Specifically, we select the following classes from FewRel: P706, P57, P22, P123, P127, P25, P17, P551, P206, P58, P40, P35, P26, P131, P937.

For MAVEN, we select Achieve, Telling, Legality, Removing, Participation, Change_sentiment, Attack, Motion, Body_movement, Becoming, Creating, Defending, Arriving, Statement, Legal_rulings, Escaping, Competition, Perception_active, Terrorism, Self_motion, Emptying, Change, Manufacturing, Hold.

## G   Complete Experimental Results

The complete experimental results on all 10 tasks are shown in Table 5.

**FewRel**

| Models | $\mathcal{T}_1$ | $\mathcal{T}_2$ | $\mathcal{T}_3$ | $\mathcal{T}_4$ | $\mathcal{T}_5$ | $\mathcal{T}_6$ | $\mathcal{T}_7$ | $\mathcal{T}_8$ | $\mathcal{T}_9$ | $\mathcal{T}_{10}$ |
|---|---|---|---|---|---|---|---|---|---|---|
| IDBR | 97.9 | 91.9 | 86.8 | 83.6 | 80.6 | 77.7 | 75.6 | 73.7 | 71.7 | 68.9 |
| KCN | 98.3 | 93.9 | 90.5 | 87.9 | 86.4 | 84.1 | 81.9 | 80.3 | 78.8 | 76.0 |
| KDRK | 98.3 | 94.1 | 91.0 | 88.3 | 86.9 | 85.3 | 82.9 | 81.6 | 80.2 | 78.0 |
| EMAR | 98.1 | 94.3 | 92.3 | 90.5 | 89.7 | 88.5 | 87.2 | 86.1 | 84.8 | 83.6 |
| RP-CRE | 97.8 | 94.7 | 92.1 | 90.3 | 89.4 | 88.0 | 87.1 | 85.8 | 84.4 | 82.8 |
| CRL | 98.2 | 94.6 | 92.5 | 90.5 | 89.4 | 87.9 | 86.9 | 85.6 | 84.5 | 83.1 |
| CRECL | 97.8 | 94.9 | 92.7 | 90.9 | 89.4 | 87.5 | 85.7 | 84.6 | 83.6 | 82.7 |
| ACA | 98.3 | 95.0 | 92.6 | 91.3 | 90.4 | 89.2 | 87.6 | 87.0 | 86.3 | 84.7 |
| CEAR | 98.1 | **95.8** | **93.6** | 91.9 | 91.1 | 89.4 | 88.1 | 86.9 | 85.6 | 84.2 |
| InfoCL | **98.3**$_{\pm0.6}$ | 95.2$_{\pm1.6}$ | 93.4$_{\pm1.3}$ | **92.1**$_{\pm1.5}$ | **91.3**$_{\pm1.4}$ | **89.7**$_{\pm1.7}$ | **88.5**$_{\pm0.3}$ | **87.7**$_{\pm1.1}$ | **86.8**$_{\pm0.6}$ | **85.4**$_{\pm0.1}$ |

**TACRED**

| Models | $\mathcal{T}_1$ | $\mathcal{T}_2$ | $\mathcal{T}_3$ | $\mathcal{T}_4$ | $\mathcal{T}_5$ | $\mathcal{T}_6$ | $\mathcal{T}_7$ | $\mathcal{T}_8$ | $\mathcal{T}_9$ | $\mathcal{T}_{10}$ |
|---|---|---|---|---|---|---|---|---|---|---|
| IDBR | 97.9 | 91.1 | 83.1 | 76.5 | 74.2 | 70.5 | 66.6 | 64.2 | 63.8 | 60.1 |
| KCN | 98.9 | 93.1 | 87.3 | 80.2 | 79.4 | 77.2 | 73.8 | 72.1 | 72.2 | 70.6 |
| KDRK | **98.9** | 93.0 | 89.1 | 80.7 | 79.0 | 77.0 | 74.6 | 72.9 | 72.1 | 70.8 |
| EMAR | 98.3 | 92.0 | 87.4 | 84.1 | 82.1 | 80.6 | 78.3 | 76.6 | 76.8 | 76.1 |
| RP-CRE | 97.5 | 92.2 | 89.1 | 84.2 | 81.7 | 81.0 | 78.1 | 76.1 | 75.0 | 75.3 |
| CRL | 97.7 | 93.2 | 89.8 | 84.7 | 84.1 | 81.3 | 80.2 | 79.1 | 79.0 | 78.0 |
| CRECL | 96.6 | 93.1 | 89.7 | 87.8 | 85.6 | 84.3 | 83.6 | 81.4 | 79.3 | 78.5 |
| ACA | 98.0 | 92.1 | 90.6 | 85.5 | 84.4 | 82.2 | 80.0 | 78.6 | 78.8 | 78.1 |
| CEAR | 97.7 | **94.3** | **92.3** | **88.4** | **86.6** | **84.5** | **82.2** | **81.1** | **80.1** | **79.1** |
| InfoCL | 96.3$_{\pm1.5}$ | 92.4$_{\pm2.1}$ | 88.9$_{\pm3.5}$ | 87.3$_{\pm2.9}$ | 83.9$_{\pm1.7}$ | 82.4$_{\pm2.5}$ | 82.0$_{\pm1.6}$ | 79.7$_{\pm0.9}$ | 78.4$_{\pm1.4}$ | 78.2$_{\pm1.7}$ |

**MAVEN**

| Models | $\mathcal{T}_1$ | $\mathcal{T}_2$ | $\mathcal{T}_3$ | $\mathcal{T}_4$ | $\mathcal{T}_5$ | $\mathcal{T}_6$ | $\mathcal{T}_7$ | $\mathcal{T}_8$ | $\mathcal{T}_9$ | $\mathcal{T}_{10}$ |
|---|---|---|---|---|---|---|---|---|---|---|
| IDBR | 96.5 | 85.3 | 79.4 | 76.3 | 74.2 | 69.8 | 67.5 | 64.4 | 60.2 | 57.3 |
| KCN | 97.2 | 87.7 | 83.2 | 80.3 | 77.9 | 75.1 | 71.9 | 68.4 | 67.7 | 64.4 |
| KDRK | 97.2 | 88.6 | 84.3 | 81.6 | 78.1 | 75.8 | 72.5 | 69.6 | 68.9 | 65.4 |
| EMAR | 97.2 | 91.4 | 88.3 | 86.1 | 83.6 | 81.2 | 79.0 | 76.8 | 75.7 | 73.2 |
| RP-CRE | 96.7 | 91.8 | 88.2 | 86.5 | 83.9 | 81.4 | 79.4 | 77.1 | 76.0 | 73.6 |
| CRL | 96.0 | 90.7 | 87.1 | 84.8 | 82.9 | 80.7 | 78.7 | 76.8 | 75.9 | 73.7 |
| CRECL | 96.9 | 91.4 | 86.9 | 84.8 | 82.4 | 80.4 | 77.5 | 75.9 | 75.1 | 73.5 |
| InfoCL | **97.5**$_{\pm1.1}$ | **92.5**$_{\pm3.8}$ | **88.4**$_{\pm3.9}$ | **86.7**$_{\pm2.8}$ | **84.6**$_{\pm2.7}$ | **82.5**$_{\pm1.9}$ | **80.0**$_{\pm1.3}$ | **78.2**$_{\pm1.2}$ | **77.1**$_{\pm0.9}$ | **75.3**$_{\pm0.3}$ |

**HWU64**

| Models | $\mathcal{T}_1$ | $\mathcal{T}_2$ | $\mathcal{T}_3$ | $\mathcal{T}_4$ | $\mathcal{T}_5$ | $\mathcal{T}_6$ | $\mathcal{T}_7$ | $\mathcal{T}_8$ | $\mathcal{T}_9$ | $\mathcal{T}_{10}$ |
|---|---|---|---|---|---|---|---|---|---|---|
| IDBR | 96.3 | 93.2 | 88.1 | 86.5 | 84.6 | 82.5 | 82.1 | 80.2 | 78.0 | 76.2 |
| KCN | 98.6 | 94.0 | 90.7 | 90.4 | 87.0 | 84.9 | 84.4 | 83.7 | 82.7 | 81.9 |
| KDRK | **98.6** | **94.5** | 91.2 | **90.4** | 87.3 | 86.0 | 85.8 | 85.1 | 82.5 | 81.4 |
| EMAR | 98.4 | 94.4 | **91.4** | 89.5 | 88.2 | 86.3 | 86.2 | 85.5 | 83.9 | 83.1 |
| RP-CRE | 97.6 | 93.7 | 90.1 | 88.6 | 86.5 | 86.3 | 85.1 | 84.5 | 83.8 | 82.7 |
| CRL | 98.2 | 92.8 | 88.8 | 86.5 | 84.1 | 82.4 | 82.8 | 83.1 | 81.3 | 81.5 |
| CRECL | 97.3 | 93.0 | 87.5 | 86.1 | 84.1 | 83.0 | 83.1 | 83.1 | 81.9 | 81.1 |
| InfoCL | 97.7$_{\pm1.3}$ | 93.6$_{\pm1.5}$ | 90.2$_{\pm1.2}$ | 90.0$_{\pm1.3}$ | **88.3**$_{\pm1.1}$ | **86.6**$_{\pm2.1}$ | **86.8**$_{\pm1.7}$ | **86.3**$_{\pm1.9}$ | **85.3**$_{\pm1.3}$ | **84.1**$_{\pm1.0}$ |

Table 5: Accuracy (%) on all observed classes after learning each task. The best results are marked in **bold**.