# OpenReview forum: "InfoCL: Alleviating Catastrophic Forgetting in Continual Text Classification from An Information Theoretic Perspective"
_EMNLP/2023/Conference — EMNLP 2023 Findings_

### Official Review · Reviewer_JheW · 2023-08-02

**Soundness:** 3

**Excitement:**

3: Ambivalent: It has merits (e.g., it reports state-of-the-art results, the idea is nice), but there are key weaknesses (e.g., it describes incremental work), and it can significantly benefit from another round of revision. However, I won't object to accepting it if my co-reviewers champion it.

**Paper Topic And Main Contributions:**

The paper focuses on continual learning for text classification in the class-incremental setting and propose the method, InfoCL, which utilizes fast-slow and current-past contrastive learning to learn more comprehensive representations and alleviate representation corruption.

**Reasons To Accept:**

1 The method is well-motivated and the analysis looks solid.
2 The experiments on the four classification tasks (fewrel, tacred, maven and hwu64) show the superior performance.

**Reasons To Reject:**

1 The proposed method is a general method that can be applied to many different classification task. However, in the paper, only four classification tasks are considered. More experiments should be added to validate its effectiveness.

**Reproducibility:**

4: Could mostly reproduce the results, but there may be some variation because of sample variance or minor variations in their interpretation of the protocol or method.

**Reviewer Confidence:**

2: Willing to defend my evaluation, but it is fairly likely that I missed some details, didn't understand some central points, or can't be sure about the novelty of the work.

---

> ### Author Rebuttal · Authors · 2023-08-26
>
> Thanks for your suggestions!
>
> As you pointed out, our method has the potential to be applied to a wide range of classification tasks. **In fact, we have thoroughly validated its effectiveness on numerous tasks.** However, due to the constraints of space in our paper, we were only able to include four representative classification tasks and conducted extensive experiments on them. These tasks were carefully selected to showcase the versatility and robustness of our method. Here we would like to explain the principles for selecting these four tasks and provide more experimental results on three additional tasks, which we hope can further demonstrate the effectiveness of our method.
>
> In this paper we mainly focus on continual text classification in a class-incremental setting, which requires the datasets to have a vast number of classes, e.g., 80 classes for FewRel and 120 classes for MAVEN. Therefore, some classification tasks which only contain 2 or 3 classes, such as sentimental analysis, are not suitable for the class-incremental setting. On the other hand, the four classification tasks we used include relation classification, event classification, and intent detection, which have already covered three typical classification scenarios and are widely used in previous work.
>
> Here we provide the results on three more tasks, CLINC150 [1], WebRED [2] and ReTACRED [3]. For all additional datasets, we simulate 10 class-incremental tasks by randomly dividing all classes of the dataset into 10 disjoint sets. The number of new classes in each task for CLINC150, WebRED and ReTACRED are 15, 10, 4 respectively. All hyperparameters of the experiments are the same as FewRel. The reported results are the average of 5 randomly sampled task sequences. These results will be added to the appendix.
>
> **As shown, our method outperforms baselines on six datasets (including three in the paper). We hope the extensive experiments can validate the effectiveness of our method.**
>
> CLINC150 [1] is an intent detection dataset with 150 intent classes. Each class has 100 instances for training and 20 instances for test. Here are the experimental results on CLINC150.
>
> | Models        | T1       | T2       | T3       | T4       | T5       | T6       | T7       | T8       | T9       | T10  |
> | ------------- | -------- | -------- | -------- | -------- | -------- | -------- | -------- | -------- | -------- | ---- |
> | RP-CRE        | 98.8     | 98.5     | 97.7     | 97.0     | 95.9     | 95.9     | 95.4     | 95.0     | 94.8     | 94.3 |
> | CRL           | **99.1** | 98.4     | 97.2     | 96.7     | 95.6     | 95.5     | 95.4     | 95.0     | 94.7     | 94.5 |
> | CRECL         | 98.4     | 97.9     | 96.3     | 96.0     | 95.3     | 94.9     | 94.8     | 94.7     | 94.4     | 94.3 |
> | InfoCL (Ours) | 99.1     | **98.8** | **98.1** | **97.5** | **96.6** | **96.2** | **96.0** | **95.6** | **95.4** | **95.1** |
>
> WebRED [2] is a severely long-tailed relation classification dataset with more than 500 classes. We use the data of the top-100 frequent classes. For each class, we randomly split the data into train and test set by 4:1. Here are the experimental results on WebRED.
>
> | Models        | T1       | T2       | T3       | T4       | T5       | T6       | T7       | T8       | T9       | T10  |
> | ------------- | -------- | -------- | -------- | -------- | -------- | -------- | -------- | -------- | -------- | ---- |
> | RP-CRE        | 98.8     | 92.5     | 85.6     | 85.3     | 80.3     | 76.2     | 72.9     | 71.1     | 66.9     | 65.4 |
> | CRL           | **98.9** | 91.7     | 83.0     | 81.4     | 76.2     | 71.8     | 68.3     | 66.3     | 63.0     | 61.7 |
> | CRECL         | 98.6     | 92.7     | 85.0     | 83.3     | 77.4     | 73.4     | 70.3     | 68.6     | 65.2     | 64.5 |
> | InfoCL (Ours) | 98.8     | **92.9** | **86.8** | **85.4** | **80.7** | **76.7** | **73.9** | **71.8** | **68.5** | **67.1** |
>
> ReTACRED [3] is a re-annotated version of TACRED which can be used to perform reliable evaluation of relation classification models. Since the dataset has 39 classes which is not a multiple of 10 (the length of the task sequence), we randomly sample a task and allocate 3 classes for it. Here are the experimental results on ReTACRED.
>
> | Models        | T1       | T2       | T3       | T4       | T5       | T6       | T7       | T8       | T9       | T10  |
> | ------------- | -------- | -------- | -------- | -------- | -------- | -------- | -------- | -------- | -------- | ---- |
> | RP-CRE        | 96.5     | 93.0     | 92.4     | 89.5     | 88.5     | 87.6     | 86.2     | 85.0     | 83.9     | 83.8 |
> | CRL           | 94.9     | 91.8     | 88.5     | 86.0     | 84.0     | 81.2     | 79.6     | 78.1     | 77.5     | 78.9 |
> | CRECL         | 96.4     | 89.9     | 87.1     | 83.0     | 82.3     | 80.6     | 82.8     | 81.7     | 81.7     | 82.7 |
> | InfoCL (Ours) | **97.1** | **94.6** | **93.4** | **91.5** | **89.7** | **88.5** | **88.0** | **86.8** | **86.0** | **85.6** |
>
> [1] An Evaluation Dataset for Intent Classification and Out-of-Scope Prediction, EMNLP 2019
>
> [2] WebRED: Effective Pretraining And Finetuning For Relation Extraction On The Web, arXiv 2102
>
> [3] Re-TACRED: Addressing Shortcomings of the TACRED Dataset, AAAI 2021

---

### Official Review · Reviewer_AarQ · 2023-08-04

**Soundness:** 3

**Excitement:**

3: Ambivalent: It has merits (e.g., it reports state-of-the-art results, the idea is nice), but there are key weaknesses (e.g., it describes incremental work), and it can significantly benefit from another round of revision. However, I won't object to accepting it if my co-reviewers champion it.

**Missing References:**

Refer to reasons to reject

**Paper Topic And Main Contributions:**

The authors propose a replay-based continual learning method for text classification problem in the class-incremental learning setting. Based on insights from information theoretic perspective, the authors explain forgetting between cross-task analogous classes and present a training approach for comprehensive representation learning. The training process consists of training the network using fast-slow contrastive learning and current-past contrastive learning with a memory buffer. The proposed method is compared with several recent CL methods in NLP and demonstrates competitive performance.

**Reasons To Accept:**

1. The information theoretic analysis is interesting
2. The experiment is thorough and shows the effectiveness of the proposed method

**Reasons To Reject:**

1. The writing can be improved for clarity. For instance, i) it's not clear why fast-slow method works especially in the information theoretic perspective, and ii) it's not clear how to implement it. Pseudo-code may be informative.
2. Similar methods have been already explored. [1] proposes mutual information maximization for holistic representation using contrastive loss. [2] proves the role of out-of-distribution of classes from future and previous tasks in class-incremental learning and proposes a method using contrastive learning.
3. Code is not provided.
4. Information about computational resources is not provided. Should provide e.g., how efficient is the proposed method, and how much GPU memory does it require?

[1] Online Continual Learning through Mutual Information Maximization. ICML 2022
[2] A Theoretical Study on Solving Continual Learning. NeurIPS 2022

I am willing to change my score if the authors address my concerns in rebuttal

**Reproducibility:**

2: Would be hard pressed to reproduce the results. The contribution depends on data that are simply not available outside the author's institution or consortium; not enough details are provided.

**Reviewer Confidence:**

3: Pretty sure, but there's a chance I missed something. Although I have a good feel for this area in general, I did not carefully check the paper's details, e.g., the math, experimental design, or novelty.

---

> ### Author Rebuttal · Authors · 2023-08-26
>
> Thank you for your helpful comments and suggestions! Please see our responses below.
>
> **Q1:** Why does the fast-slow method work, especially in the information theoretic perspective? How to implement it?
>
> **A1:** The fast-slow contrastive method works since it can maximize the mutual information between the input sentences and the representations $I(X_i;Z_i)$. First, it uses the InfoNCE loss which can directly maximize $I(X_i;Z_i)$ [3]. Second, according to the information bottleneck theory [4], in the early phase of optimization, e.g., the first several epochs, $I(X_i, Z_i)$ is larger. As a result, the representations $Z_i$ from the slowly-updated branch will preserve more information about the input sentences. The fast-slow contrast can "distill" these information from the slow branch to fast branch to learn more comprehensive representations.
>
> The code of the fast-slow method is provided in Answer 3. Due to space limitation, we did not provide the pseudo code and would like to put it in the appendix in the new version. That will make the method more readable. Thanks again for your suggestion!
>
> **Q2:** What are the differences with the related work [1] and [2]?
>
> **A2:** Our work, in comparison to OCM [1], shares a similar motivation of learning more comprehensive/holistic representations. We would like to provide clarifications on several key differences between our work and [1]:
>
> 1. First, [1] focuses on online continual learning on image classification, which is different with our scenario.
> 2. Second, for the first time, we provide a formal analysis of catastrophic forgetting from the perspective of the information bottleneck. This analysis sets us apart from [1].
> 3. Finally, the contrastive learning in OCM relies on data augmentation of the image to ensure performance (According to the ablation study in [1], without the random-resized-crop augmentation, the performance will drop drastically). In contrast, we design fast-slow and current-past contrastive learning to get better representations. This contrastive learning design is also a distinguishing feature of our work.
>
> Regarding [2], it employs the out-of-distribution method CSI [5], which uses contrastive learning to conduct OOD detection, instead of helping the CL model learn more comprehensive representations.
>
> **Q3:** Code is not provided.
>
> **A3:** Our code and continual learning framework will be publicly released upon acceptance. Here is an anonymous version of the code of our method: https://anonymous.4open.science/r/InfoCL_EMNLP-DAD6/
>
> **Q4:** Information about computational resources is not provided.
>
> **A4:** We have recorded the computational resources used during the experiments. These results are listed below for your reference.
>
> All experiments are conducted on a single NVIDIA A800 GPU. The execution time (minutes per. round) of our method and baselines are as follows:
>
> |        | FewRel   | TACRED   | MAVEN    | HWU64   |
> | ------ | -------- | -------- | -------- | ------- |
> | RP-CRE | 34.1     | 10.2     | 85.2     | 8.4     |
> | CRL    | 31.0     | **10.0** | **71.5** | **6.8** |
> | CRECL  | 40.3     | 12.0     | 79.6     | 8.0     |
> | ACA    | **28.5** | 12.6     | -        | -       |
> | CEAR   | 48.5     | 13.4     | -        | -       |
> | InfoCL | 29.2     | 11.6     | 78.4     | 7.0     |
>
> Although we utilize two contrastive learning losses and an adversarial mechanism, our method remains remarkable efficiency. This can be attributed to the parallel execution and gradient truncation in the slow branch of fast-slow contrast and the past branch in current-past contrast.
>
> The GPU memory utilization depends on the batch size and the sentence length of the dataset. For FewRel, MAVEN and HWU64, the batch size is 32. For TACRED, the batch size is 16. The GPU memory utilization is: 4138MB for FewRel, 8926MB for TACRED, 14958MB for MAVEN, 3692MB for HWU64. These details will be put into the new version according to your suggestions.
>
> [1] Online Continual Learning through Mutual Information Maximization. ICML 2022
>
> [2] A Theoretical Study on Solving Continual Learning. NeurIPS 2022
>
> [3] On Variational Bounds of Mutual Information. ICML 2019
>
> [4] Opening the Black Box of Deep Neural Networks via Information. ICLR 2018
>
> [5] Csi: Novelty detection via contrastive learning on distributionally shifted instances. NeuIPS 2020

---

### Official Review · Reviewer_EZuw · 2023-08-06

**Soundness:** 3

**Excitement:**

3: Ambivalent: It has merits (e.g., it reports state-of-the-art results, the idea is nice), but there are key weaknesses (e.g., it describes incremental work), and it can significantly benefit from another round of revision. However, I won't object to accepting it if my co-reviewers champion it.

**Paper Topic And Main Contributions:**

This paper focuses on enable the model learn more sufficient representations to solve the information bottleneck issues,  they propose the fast-slow and current-past contrastive learning to learn from new tasks and old tasks, which maximum the mutual information. Experimenta result prove the effectiveness of this approach.

**Questions For The Authors:**

Please see "Reasons To Reject"

**Reasons To Accept:**

1. This paper is well organized, the description of the representation bias, methodology is relative clearly to follow, also the perspective of the representation bias instead of sampling bias (e.g., coreset) is interesting.

2. Compared with existing approach that also used contrastive learning appraoch (e.g., Co2L), this paper focused on fast and slow learning mechanism perspective to solve the bias problem, which is relative novel idea. I also see the phenomenon that different learning rate on encoder and linear layer for feature representation is useful, here the author could explain it more reasonable with more specfic work.

3. The experimental result show the obvious improvement compared with existing SOTA, also the relative ablation study is specific and convinced.

**Reasons To Reject:**

1. Some relative works[1,2] should also be discussed, which also applied contrastive learning (infoNCE) on continual learning to solve the bias or generalization work. The author applied the InfoNCE for optimize the fast and slow learning, so the discussion with other work that also applied the InforNCE should be considered.

2. Some unclear description in the paper, e.g., in Sec 5.5, what's the motivation that used the Adversarial Memory Augmentation here for augmentation, instead of other approach in the NLP? In the experimental setting, what's the influence from the size of memory buffer? Do we have some strategy on improving the sampling strategy on memory buffer?


[1]. Co2L: Contrastive Continual Learning. CVPR 2021
[2]. CLASSIC: Continual and Contrastive Learning of Aspect Sentiment Classification Tasks. EMNLP 2012

**Reproducibility:**

3: Could reproduce the results with some difficulty. The settings of parameters are underspecified or subjectively determined; the training/evaluation data are not widely available.

**Reviewer Confidence:**

3: Pretty sure, but there's a chance I missed something. Although I have a good feel for this area in general, I did not carefully check the paper's details, e.g., the math, experimental design, or novelty.

---

> ### Author Rebuttal · Authors · 2023-08-26
>
> Thanks for your insightful comments! Please see our reponses below.
>
> **Q1:** Some related work should also be discussed.
>
> **A1:** Thank you for suggesting the inclusion of a discussion on other related work.
> In our work, we have already compared our approach with CRL [2], which utilizes the same method as Co2L [1]. We will add a reference to Co2L [1] in our paper, as you suggested. Additionally, we would like to highlight two significant differences between our work and Co2L. First, we formally analyze the representation bias problem from an information bottleneck perspective and propose that maximizing $I(X_i;Z_i)$ can effectively mitigate this bias. Second, while Co2L employs a vanilla supervised contrastive loss for representation learning, we introduce two novel contrastive learning techniques: fast-slow and current-past contrastive learning. Our method not only enables the model to acquire more comprehensive representations but also improves its ability to retain previously learned knowledge. As shown in Table 2, our method outperforms CRL/Co2L on four datasets.
>
> Regarding CLASSIC, it is specifically designed for the domain-incremental scenario, where its primary goal is to facilitate knowledge transfer across tasks. This objective is distinct from our class-incremental scenario, and thus, the use of InfoNCE in CLASSIC is not directly comparable to our work.
>
>
> **Q2:** The motivation of Adversarial Memory Augmentation is unclear.
>
> **A2:** In replay-based CL methods, only a few instances of each class will be stored into the memory, which are then replayed during subsequent tasks. Therefore, the performance of replay is hindered by the overfitting issue. Adversarial data augmentation has emerged as a powerful technique for reducing overfitting [3, 4, 5]. Inspired by this, we incorporate adversarial memory augmentation to enhance the performance of current-past contrastive learning in the memory replay stage. We will improve the description according to your advice.
>
> **Q3:** What is the influence of the size of memory buffer?
>
> **A3:** The size of memory buffer is a trade-off between model performance and storage budget. With a larger memory, the model will have a better performance but it needs more storage space. The influence of memory size on model performance is often used to showcase the robustness of different methods [2, 6, 7]. We have conducted experiments and discussed the influence of memory size in Section 7.4.
>
>
> **Q4:** Do we have some strategy for improving the sampling strategy of memory buffer?
>
> **A4:** The sampling strategy is an important aspect for the replay-based CL methods. We use K-means based sampling strategy, which has demonstrated effectiveness in previous studies [2, 6, 7]. However, the primary focus of our paper is to analyze and address the issue of analogous class confusion from a representation learning perspective. As a result, we adopt the memory sampling strategy employed in prior work to ensure a fair comparison. In future work, we will explore more advanced sampling strategies to mitigate catastrophic forgetting.
>
>
> Thanks again for the useful comments. We will continue to improve our paper according to your suggestions.
>
> [1] Co2L: Contrastive Continual Learning, CVPR 2021
>
> [2] Consistent Representation Learning for Continual Relation Extraction, ACL 2022
>
> [3] Adversarial Attacks on Deep-learning Models in Natural Language Processing: A Survey. arXiv
>
> [4] Adversarial Training for Free! NeuIPS 2019
>
> [5] FreeLB: Enhanced Adversarial Training for Natural Language Understanding, ICLR 2020
>
> [6] Refining Sample Embeddings with Relation Prototypes to Enhance Continual Relation Extraction, ACL 2021
>
> [7] Learning Robust Representations for Continual Relation Extraction via Adversarial Class Augmentation, EMNLP 2022

---

### Meta-Review · Area_Chair_gYH3 · 2023-09-17

**Recommendation:** 4

**Metareview:**

This paper proposes a method using fast-slow contrastive learning with a memory buffer for continual learning.  Experimenta result prove the effectiveness of this approach.

All reviewers find the information theoretic analysis insightful and interesting. The experimental results (with the additional results provided in response) are adequate.

The discussion and writing could be improved (e.g. motivation of fast-slow contrastive learning). The authors may want to discuss related work (e.g. Co2L, OCM) and incorporate the response. We highly encourage authors add additional results provided in response.

---

### Decision · Program_Chairs · 2023-10-07

**Decision:**

Accept-Findings

**Comment:**

This paper proposes a method using fast-slow contrastive learning with a memory buffer for continual learning.  Experimenta result prove the effectiveness of this approach.

All reviewers find the information theoretic analysis insightful and interesting. The experimental results (with the additional results provided in response) are adequate.

The discussion and writing could be improved (e.g. motivation of fast-slow contrastive learning). The authors may want to discuss related work (e.g. Co2L, OCM) and incorporate the response. We highly encourage authors add additional results provided in response.